# Evaluation of Low-Complexity Adaptive Full Direct-State Kalman Filter for Robust GNSS Tracking [note 1]

**DOI:** 10.3390/s23073658

**Published:** 2023-03-31

**Authors:** Iñigo Cortés, Johannes Rossouw van der Merwe, Elena Simona Lohan, Jari Nurmi, Wolfgang Felber

**Affiliations:** 1Satellite Based Positioning Systems Department, Fraunhofer IIS, Nordostpark 84, 90411 Nuremberg, Germany; 2Electrical Engineering, Tampere University, 33014 Tampere, Finland; 3Focal Point Positioning, Cambridge CB4 3NP, UK

**Keywords:** global navigation satellite system (GNSS), full direct-state Kalman filter (DSKF), lookup table direct-state Kalman filter (LUT-DSKF), loop-bandwidth control algorithm (LBCA), adaptive tracking techniques

## Abstract

This paper evaluates the implementation of a low-complexity adaptive full direct-state
Kalman filter (DSKF) for robust tracking of global navigation satellite system (GNSS) signals. The full DSKF includes frequency locked loop (FLL), delay locked loop (DLL), and phase locked loop (PLL) tracking schemes. The DSKF implementation in real-time applications requires a high computational cost. Additionally, the DSKF performance decays in time-varying scenarios where the statistical distribution of the measurements changes due to noise, signal dynamics, multi-path, and non-line-of-sight effects. This study derives the full lookup table (LUT)-DSKF: a simplified full DSKF considering the steady-state convergence of the Kalman gain. Moreover, an extended version of the loop-bandwidth control algorithm (LBCA) is presented to adapt the response time of the full LUT-DSKF. This adaptive tracking technique aims to increase the synchronization robustness in time-varying scenarios. The proposed tracking architecture is implemented in an GNSS hardware receiver with an open software interface. Different configurations of the adaptive full LUT-DSKF are evaluated in simulated scenarios with different dynamics and noise cases for each implementation. The results confirm that the LBCA used in the FLL-assisted-PLL (FAP) is essential to maintain a position, velocity, and time (PVT) fix in high dynamics.

## 1. Introduction

Global navigation satellite system (GNSS) receivers require reliable synchronization with incoming GNSS signals to achieve a continuous position, velocity, and time (PVT) solution [1]. The synchronization process consists of two stages: acquisition and tracking. Acquisition coarsely estimates the code phase and the carrier Doppler of received GNSS signals. The tracking stage refines these synchronization parameters and includes the fine estimation of the carrier phase. A successful synchronization permits the decoding of the navigation message and the estimation of the pseudo-range and pseudo-range rate, which finally leads to the PVT calculation [2,3].

The carrier phase ϕ, carrier Doppler *f*, and code phase τ are the main parameters the GNSS receiver synchronizes with. Standard tracking techniques use scalar tracking loops (STLs) in the tracking stage. This tracking scheme synchronizes with a single synchronization parameter of a GNSS signal at a time [1,3]. Thus, a tracking channel includes three STLs: phase locked loop (PLL), frequency locked loop (FLL), and delay locked loop (DLL). The STL contains a correlator, a discriminator, a loop filter, and a numerically controlled oscillator (NCO) [4,5]. The configuration parameters of the STL include the discriminator type, the loop bandwidth *B*, the integration time $\tau_{\mathrm{int}}$, the order *p*, and the correlator spacing $\Delta_{s}$. These parameters determine the performance and robustness against noise and signal dynamics. The well-known trade-off between noise filtering capabilities and signal dynamics resistance is the main challenge of fix-configured STLs. In particular, this problem is aggravated in time-varying scenarios. These scenarios are characterized by different realizations of signal dynamics, noise, and fading effects that lead to challenges regarding synchronization capability [1]. For instance, a high-order STL with wide loop bandwidth and short integration time is adequate to track rapidly changing parameters. In contrast, a low-order STL with narrow loop bandwidth and long integration time is preferable to track noisy parameters. Therefore, a fixed configuration of the STL is a sub-optimal solution for time-varying scenarios.

Carrier-phase continuity in mobile devices is fundamental to achieving decimeter-level positioning through real-time kinematic (RTK) [6] or precise point positioning (PPP) [7,8]. However, smartphones use ultra-low-cost GNSS chipsets and low-gain antennas leading to poor GNSS observations [9], challenging carrier phase continuity and, in turn, decimeter-level positioning. Moreover, GNSS observations are highly affected by multipath, particularly in dense urban scenarios [10]. Additionally, vehicular scenarios usually experience short outages, where GNSS signals can be shortly blocked by residential buildings, overpasses, or short tunnels, interrupting the GNSS observations. Therefore, searching for a robust tracking technique that maintains the carrier phase continuity under these scenarios is highly necessary.

Size, weight, and power (SWAP) are key metrics for GNSS mass-market chip manufacturing. In particular, power consumption is a relevant topic in mobile devices, and several power-saving techniques have been proposed [11,12]. When the GNSS receiver loses the synchronization of the GNSS signals, the re-acquisition is performed, returning to the acquisition stage. Since acquisition is a power-consuming process, a robust tracking architecture can avoid re-acquisition by not losing the lock of the GNSS signal, decreasing the power consumption significantly.

GNSS receivers like the GOOSE©platform [13] partially implement the tracking stage in hardware (correlators and NCO ) and software (discriminators and loop filters). These receivers try to close the loop of all the tracking channels before a new correlation is performed. A low time complexity of the software implementation is essential to close the loop on time, avoiding synchronization failures. Furthermore, the lower the time complexity, the more tracking channels the GNSS receiver can manage. Hence, a low-complexity robust tracking architecture is critical to achieving a low time complexity and, in turn, more tracking channels.

The Kalman filter (KF) is an optimal infinite impulse response (IIR) estimator under the assumption of linear Gaussian error statistics [14,15,16]. Knowledge of the process noise covariance Q and the measurement noise covariance R allows the KF to adapt its coefficients optimally, achieving the minimum mean square error (MMSE) [17]. There are several KF implementation methods in STLs [18] grouped into error-state Kalman-filter (ESKF) and direct-state Kalman filter (DSKF) [19]. The former replaces the loop filter of the STL with a KF [20,21,22,23], whereas the latter considers the whole STL as part of the KF [24,25,26,27,28]. The implementation of the DSKF is straightforward due to the relation between the STL’s coefficients and the DSKF’s Kalman gains [24].

The MMSE is only achieved if prior knowledge of Q and R is available or if these are accurately estimated [17]. If this is not the case, the KF converges to a suboptimal solution [29]. Hence, for time-varying scenarios in which Q and R continuously change, the DSKF and STL share the same challenge in synchronization capability.

There has been significant research towards robust tracking solutions to solve this problem [30]. However, there are still ample opportunities to find the best technique in terms of performance and complexity [25,31]. Adaptive tracking methods can improve the tracking performance in time-varying scenarios. Different methods to estimate the noise covariances of the KF have been summarized in a review study [32]. One solution can be to implement a moving average filter to estimate Q and R and, consequently, adapt the response time of the KF optimally [33]. Moreover, it is possible to implement a carrierto-noise density ratio (*C*/*N*_0_)-based DSKF, in which R depends on the variance of the STL discriminator output [27]. Q can also be adapted according to the dynamic stress error [28]. Recent research implements the loop-bandwidth control algorithm (LBCA)-based DSKF for the PLL [24]. The LBCA performs a loop bandwidth-dependent weighted difference between estimated noise and estimated dynamics of the discriminator output [34]. This algorithm updates the loop bandwidth and, in turn, Q, based on the steady-state relationship.

Despite the tracking performance advantage of the KF, its implementation in real-time applications requires a high computational cost compared to the STL. Therefore, efficient low-complexity methods have been studied [23,25]. The complexity of the ESKF can be reduced by taking advantage of the Kalman gain convergence in the steady state [23]. The same can be done for the DSKF, leading to the so-called lookup table (LUT)-DSKF [25]. The implementation of an LBCA-based LUT-DSKF in a PLL tracking scheme has been presented recently [25]. The ratio between the steady-state process variance and the measurement variance provides a one-to-one relationship between the steady-state Kalman gains and the loop bandwidth. Hence, the LBCA can adapt the loop bandwidth to, in turn, adapt the steady-state Kalman gains.

Aiding the FLL in the PLL can significantly improve the robustness against high signal dynamics [35]. Recent research implements an LBCA-based FLL-assisted-PLL (FAP) architecture [36]. This adaptive tracking architecture consists of two independent LBCAs to adapt the bandwidths of a second-order FLL and a third-order PLL. Despite the promising results, extensive tuning was required to find the optimal weighting functions for each LBCA. Furthermore, the second LBCA doubles the complexity.

Figure 1 shows the relation of relevant research on LBCA-based techniques. First, the implementation of the LBCA in tracking schemes with only one measurement has been studied. From the STL [31,34,37,38,39] to more advance tracking schemes such as the DSKF [24] and the LUT-DSKF [25]. Second, the research has been recently expanded by implementing the LBCA in tracking schemes with two measurements. Recent research implements an adaptive LUT-DSKF in a FAP tracking architecture [26]. The derivation of the discrete algebraic Riccati equation (DARE) of this tracking architecture presents an inter-dependency between FLL and PLL coefficients. Only one LBCA can adapt the LUT-DSKF’s response time based on the found inter-dependency. This architecture has been also evaluated under simulated controlled fading scenarios [40]. Furthermore, recent studies show the tracking performance of the LBCA under simulated moon exploration missions [41].

This research expands a conference paper [26]. First, the code phase estimation is included in the DSKF leading to the full DSKF. Second, the DARE derivation of the full DSKF obtains the full LUT-DSKF: a low-complexity tracking structure that uses the steady-state Kalman gains. The derivation shows the same steady-state coefficients that update the frequency Doppler and the carrier phase for the full LUT-DSKF and the LUT-DSKF in the FAP tracking scheme. This study also presents the full LUT-DSKF steady-state coefficients that update the code phase and remarks on the impact of the PLL-assisted-DLL (PAD) on the coefficients. Third, the LBCA is expanded to adapt the full LUT-DSKF. The same LBCA as in the conference paper is used to adapt the response time of the FAP in the full LUT-DSKF. Additionally, this research presents a second LBCA to update the DLL’s response time. Fourth, instead of evaluating the tracking performance of a particular satellite vehicle (SV), as presented in the conference paper [26], the carrier and code system performance metrics are selected. These metrics consider all the visible tracked SVs and indicate an overall performance of the tracking architectures under test.

This paper shows the adaptive LUT-DSKF, a compact representation of a robust single-frequency adaptive tracking architecture considering all the primary synchronization parameters. This architecture is implemented in the tracking stage of a GOOSE© receiver [13]. The system performance of different adaptive full LUT-DSKF configurations are evaluated under simulated scenarios with different dynamics and noise levels.

The rest of the paper is organized as follows. Section 2 describes the full DSKF. The analysis of this tracking scheme in the analog and discrete domain is performed, presenting the system and measurement model, the state space model (SSM) representation, and the steady-state convergence. Section 3 shows the architecture of the integration of the LBCA in the full LUT-DSKF. Section 4 presents the experimental setup and Section 5 the achieved results. Finally, Section 6 concludes and indicates future work.

## 2. Full Direct-State Kalman Filter in Tracking Stage

This section describes the full DSKF tracking scheme of a GNSS receiver. First, the full DSKF is analyzed in the analog domain. The system and measurement models, the SSM representation, and the derivation of the continuous domain algebraic Riccati equation (CARE) is shown. Second, the full DSKF in the discrete domain is presented. As in the analog domain, the system and measurement models, the SSM, and the DARE are derived. Finally, the linear model of the steady-state full DSKF, the LUT-DSKF is shown.

### 2.1. Analog Domain

Assuming a Brownian motion model for the angular acceleration state and the code phase [42], the system model is represented as:(1)$$\begin{array}{cc} \underset{\dot{x}\left(t\right)}{\underset{︸}{\left[\begin{array}{c} \dot{\tau }\left(t\right)\\ \dot{\phi }\left(t\right)\\ \dot{f}\left(t\right)\\ \dot{a}\left(t\right) \end{array}\right]}} & =\underset{A}{\underset{︸}{\left[\begin{array}{cccc} 0 & 0 & \upsilon & 0\\ 0 & 0 & 1 & 0\\ 0 & 0 & 0 & 1\\ 0 & 0 & 0 & 0 \end{array}\right]}}\underset{x(t)}{\underset{︸}{\left[\begin{array}{c} \tau (t)\\ \phi (t)\\ f(t)\\ a(t) \end{array}\right]}}+\underset{w(t)}{\underset{︸}{\left[\begin{array}{c} w_{\tau }\left(t\right)\\ 0\\ 0\\ w_{a}\left(t\right) \end{array}\right]}} \end{array}$$
where *t* is the time index, x is the state vector composed of the code phase τ, the carrier phase ϕ, the carrier Doppler *f*, and the angular acceleration *a*. The the rate of the state vector $\dot{x}\left(t\right)$ consists of the respective deviates (i.e., rates) $\left\{\dot{\tau },\dot{\phi },\dot{f},\dot{a}\right\}$. A is the state transition matrix, and w is the process noise vector. w consists of the zero-mean Gaussian distributed perturbations that suffer the code phase in chips/s and angular acceleration in $\mathrm{cycles}/s^{3}$, denoted as $w_{\tau }$ and $w_{a}$. The parameter υ is a scaling factor that determines the aiding of the carrier Doppler state *f* into the code phase rate $\dot{\tau }$. This parameter changes if PAD is enabled or disabled. It is defined as:(2)$$\upsilon =\left\{\begin{array}{cc} 0 & \mathrm{if}\;\mathrm{PAD}\;\mathrm{disabled}\\ \frac{f_{c}}{f_{r}} & \mathrm{if}\;\mathrm{PAD}\;\mathrm{enabled} \end{array}\right.$$
where $f_{c}$ is the chipping rate in chips/s and $f_{r}$ is the carrier frequency of the GNSS signal in Hz.

The process variance of the analog system model Q is represented as:(3)$$\begin{array}{c} Q=E\left[ww^{T}\right]=\left[\begin{array}{cccc} q_{\tau } & 0 & 0 & 0\\ 0 & 0 & 0 & 0\\ 0 & 0 & 0 & 0\\ 0 & 0 & 0 & q_{a} \end{array}\right] \end{array}$$
where E[·] is the average operation, and $q_{\tau }$ and $q_{a}$ are the variances of the random processes $w_{\tau }$ in ${\mathrm{chips}}^{2}/s^{2}$ and $w_{a}$ in ${\mathrm{cycles}}^{2}/s^{6}$.

The full DSKF has three measurements from the main synchronization parameters: the code phase $z_{\tau }$, the carrier phase $z_{\phi }$, and the carrier frequency $z_{f}$. The relation between measurements and states is:(4)$$\begin{array}{c} \underset{z(t)}{\underset{︸}{\left[\begin{array}{c} z_{\tau }\left(t\right)\\ z_{\phi }\left(t\right)\\ z_{f}\left(t\right) \end{array}\right]}}=\underset{H}{\underset{︸}{\left[\begin{array}{cccc} 1 & 0 & 0 & 0\\ 0 & 1 & 0 & 0\\ 0 & 0 & 1 & 0 \end{array}\right]}}x\left(t\right)+\underset{v(t)}{\underset{︸}{\left[\begin{array}{c} v_{\tau }\left(t\right)\\ v_{\phi }\left(t\right)\\ v_{f}\left(t\right) \end{array}\right]}} \end{array}$$
where z is the measurement vector, H is the measurement matrix, and v is the measurement noise vector. The random variables (RVs)$\left\{v_{\tau },v_{\phi },v_{f}\right\}$ represent the zero-mean Gaussian distributed noise of $\left\{z_{\tau },z_{\phi },z_{f}\right\}$.

The measurement noise covariance matrix R is:(5)$$\begin{array}{cc} R & =E\left[zz^{T}\right]=\left[\begin{array}{ccc} R_{\tau } & 0 & 0\\ 0 & R_{\phi } & 0\\ 0 & 0 & R_{f} \end{array}\right] \end{array}$$
where $\left\{R_{\tau },R_{\phi },R_{f}\right\}$ are the variances of $\left\{v_{\tau },v_{\phi },v_{f}\right\}$.

The continuous SSM is derived from Equations (Equation 1) and (Equation 4):(6)x˙(t)=Ax(t)+ν3α3β3ν2α2β2ν1α1β1ν0α0β0︸Kδτ(t)δϕ(t)δf(t)ν3α3β3ν2α2β2ν1α1β1ν0α0β0︸δz(t)
(7)$$\begin{array}{c} \underset{\hat{z}\left(t\right)}{\underset{︸}{\left[\begin{array}{c} {\hat{z}}_{\tau }\left(t\right)\\ {\hat{z}}_{\phi }\left(t\right)\\ {\hat{z}}_{f}\left(t\right) \end{array}\right]}}=H\;x\left(t\right) \end{array}$$
where K is the coefficient matrix, also known as the Kalman gain matrix. $\hat{z}$ is the predicted measurement that includes the estimated code phase ${\hat{z}}_{\tau }$, the estimated carrier phase ${\hat{z}}_{\phi }$ and the estimated carrier Doppler ${\hat{z}}_{f}$. The innovation vector δz is represented as the difference between the measurement z and the estimated measurement $\hat{z}$:(8)$$\begin{array}{c} \delta z\left(t\right)=z\left(t\right)-\hat{z}\left(t\right) \end{array}$$

The presented SSM in Equations (Equation 6) and (Equation 7) is equivalent to a Kalman-Bucy filter [43]. In the steady-state region, the error covariance matrix P converges to a steady-state value, denoted as $P_{\mathrm{ss}}$ given a constant Q and R. If the process and measurement variance matrices are known, the trace of $P_{\mathrm{ss}}$ represents the MMSE. $P_{\mathrm{ss}}$ is calculated solving the CARE [44,45]:(9)$$\begin{array}{cc} 0 & =AP_{\mathrm{ss}}+P_{\mathrm{ss}}A^{T}+Q-P_{\mathrm{ss}}H^{T}R^{-1}H^{T}P_{\mathrm{ss}} \end{array}$$

The following assumption facilitates the CARE solution [23]:(10)$$\begin{array}{c} R_{i,j}\gg {\left(H\;P_{\mathrm{ss}}\;H^{T}\right)}_{i,j}\;\forall i,j=1,2 \end{array}$$
(11)$$\begin{array}{c} R_{\tau }\gg R_{f}\upsilon^{2}\gg R_{\phi }\upsilon^{2} \end{array}$$

The approximated positive-definite solution of the CARE gives the steady-state value of the error covariance matrix $P_{\mathrm{ss}}$.
(12)$$\begin{array}{c} P_{\mathrm{ss}}=\left[\begin{array}{cccc} q_{\tau }^{1/2}R_{\tau }^{1/2} & \upsilon 2q_{a}^{1/6}R_{\phi }^{5/6} & \upsilon 2q_{a}^{1/3}R_{\phi }^{2/3} & \upsilon q_{a}^{1/2}R_{\phi }^{1/2}\\ \mathrm{sym}. & 2q_{a}^{1/6}R_{\phi }^{5/6} & 2q_{a}^{1/3}R_{\phi }^{2/3} & q_{a}^{1/2}R_{\phi }^{1/2}\\ \mathrm{sym}. & \mathrm{sym}. & 3q_{a}^{1/2}R_{\phi }^{1/2} & 2q_{a}^{2/3}R_{\phi }^{1/3}\\ \mathrm{sym}. & \mathrm{sym}. & \mathrm{sym}. & 2q_{a}^{5/6}R_{\phi }^{1/6} \end{array}\right] \end{array}$$
where sym. is the abbreviation of symmetrical.

The steady-state Kalman gain $K_{\mathrm{ss}}$ is derived based on the calculated $P_{\mathrm{ss}}$ in Equation (Equation 12):(13)$$\begin{array}{cc} K_{\mathrm{ss}} & =P_{\mathrm{ss}}H^{T}R^{-1}=\left[\begin{array}{ccc} \nu_{{\mathrm{ss}}_{3}} & \alpha_{{\mathrm{ss}}_{3}} & \beta_{{\mathrm{ss}}_{3}}\\ \nu_{{\mathrm{ss}}_{2}} & \alpha_{{\mathrm{ss}}_{2}} & \beta_{{\mathrm{ss}}_{2}}\\ \nu_{{\mathrm{ss}}_{1}} & \alpha_{{\mathrm{ss}}_{1}} & \beta_{{\mathrm{ss}}_{1}}\\ \nu_{{\mathrm{ss}}_{0}} & \alpha_{{\mathrm{ss}}_{0}} & \beta_{{\mathrm{ss}}_{0}} \end{array}\right] \end{array}$$(14)$$\begin{array}{cc} \; & =\left[\begin{array}{ccc} \kappa & 2\upsilon \gamma & 2\upsilon \gamma^{2}\frac{R_{\phi }}{R_{f}}\\ 2\upsilon \gamma \frac{R_{\phi }}{R_{\tau }} & 2\gamma & 2\gamma^{2}\frac{R_{\phi }}{R_{f}}\\ 2\upsilon \gamma^{2}\frac{R_{\phi }}{R_{\tau }} & 2\gamma^{2} & 3\gamma^{3}\frac{R_{\phi }}{R_{f}}\\ \upsilon \gamma^{3}\frac{R_{\phi }}{R_{\tau }} & \gamma^{3} & 2\gamma^{4}\frac{R_{\phi }}{R_{f}} \end{array}\right]\approx \left[\begin{array}{ccc} \kappa & 2\upsilon \gamma & 2\upsilon \gamma^{2}\frac{R_{\phi }}{R_{f}}\\ 0 & 2\gamma & 2\gamma^{2}\frac{R_{\phi }}{R_{f}}\\ 0 & 2\gamma^{2} & 3\gamma^{3}\frac{R_{\phi }}{R_{f}}\\ 0 & \gamma^{3} & 2\gamma^{4}\frac{R_{\phi }}{R_{f}} \end{array}\right] \end{array}$$
where γ is the ratio ${\left(q_{a}/R_{\phi }\right)}^{1/6}$ and κ is the ratio ${\left(q_{\tau }/R_{\tau }\right)}^{1/2}$ in Hertz. The variables γ and κ simplify the natural formulation to improve readability. Considering the assumption in Equation (Equation 11), the steady-state coefficients of the DLL $\left\{\nu_{{\mathrm{ss}}_{0}},\nu_{{\mathrm{ss}}_{1}},\nu_{{\mathrm{ss}}_{2}}\right\}$ are approximated to zero. The steady-state coefficients of the PLL $\left\{\alpha_{{\mathrm{ss}}_{0}},\alpha_{{\mathrm{ss}}_{1}},\alpha_{{\mathrm{ss}}_{2}},\alpha_{{\mathrm{ss}}_{3}}\right\}$ and the FLL $\left\{\beta_{{\mathrm{ss}}_{0}},\beta_{{\mathrm{ss}}_{1}},\beta_{{\mathrm{ss}}_{2}},\beta_{{\mathrm{ss}}_{3}}\right\}$ depend on γ, whereas the remainder coefficient of the DLL $\nu_{{\mathrm{ss}}_{3}}$ is dependent on κ. These two parameters {γ,κ} determine the time of response of the full LUT-DSKF. Furthermore, the dependency of $\left\{\alpha_{{\mathrm{ss}}_{3}},\beta_{{\mathrm{ss}}_{3}}\right\}$ with υ indicates that, if PAD is enabled, the carrier phase error δϕ and the code phase error δτ will have an influence on the code phase rate estimation $\dot{\tau }$ (see Equation (Equation 6)).

### 2.2. Digital Domain

Based on the backward Euler transform (BET) [46,47], the discrete system model of the full DSKF is represented as:(15)$$\begin{array}{cc} \underset{x[n]}{\underset{︸}{\left[\begin{array}{c} \tau [n]\\ \phi [n]\\ f[n]\\ a[n] \end{array}\right]}} & =\underset{A_{d}}{\underset{︸}{\left[\begin{array}{cccc} 1 & 0 & \upsilon \tau_{\mathrm{int}} & \upsilon \tau_{\mathrm{int}}^{2}\\ 0 & 1 & \tau_{\mathrm{int}} & \tau_{\mathrm{int}}^{2}\\ 0 & 0 & 1 & \tau_{\mathrm{int}}\\ 0 & 0 & 0 & 1 \end{array}\right]}}\underset{x[n-1]}{\underset{︸}{\left[\begin{array}{c} \tau [n-1]\\ \phi [n-1]\\ f[n-1]\\ a[n-1] \end{array}\right]}}+\underset{G}{\underset{︸}{\left[\begin{array}{cccc} \tau_{\mathrm{int}} & 0 & \upsilon \tau_{\mathrm{int}}^{2} & \upsilon \tau_{\mathrm{int}}^{3}\\ 0 & \tau_{\mathrm{int}} & \tau_{\mathrm{int}}^{2} & \tau_{\mathrm{int}}^{3}\\ 0 & 0 & \tau_{\mathrm{int}} & \tau_{\mathrm{int}}^{2}\\ 0 & 0 & 0 & \tau_{\mathrm{int}} \end{array}\right]}}\underset{w[n]}{\underset{︸}{\left[\begin{array}{c} w_{\tau }\left[n\right]\\ 0\\ 0\\ w_{a}\left[n\right] \end{array}\right]}} \end{array}$$
where *n* is the sample index, $A_{d}$ is the discrete state matrix, $\tau_{\mathrm{int}}$ is the integration time, and the term Gw determines the discrete process noise that drives the signal dynamics.

The discrete process covariance matrix $Q_{d}$ is defined as:(16)$$\begin{array}{cc} Q_{d}\; & =G\;E\left[ww^{T}\right]\;G^{T}=q_{\tau }\left[\begin{array}{cccc} \tau_{\mathrm{int}}^{2} & 0 & 0 & 0\\ 0 & 0 & 0 & 0\\ 0 & 0 & 0 & 0\\ 0 & 0 & 0 & 0 \end{array}\right]+q_{a}\left[\begin{array}{cccc} \upsilon^{2}\tau_{\mathrm{int}}^{6} & \upsilon \tau_{\mathrm{int}}^{6} & \upsilon \tau_{\mathrm{int}}^{5} & \upsilon \tau_{\mathrm{int}}^{4}\\ \upsilon \tau_{\mathrm{int}}^{6} & \tau_{\mathrm{int}}^{6} & \tau_{\mathrm{int}}^{5} & \tau_{\mathrm{int}}^{4}\\ \upsilon \tau_{\mathrm{int}}^{5} & \tau_{\mathrm{int}}^{5} & \tau_{\mathrm{int}}^{4} & \tau_{\mathrm{int}}^{3}\\ \upsilon \tau_{\mathrm{int}}^{4} & \tau_{\mathrm{int}}^{4} & \tau_{\mathrm{int}}^{3} & \tau_{\mathrm{int}}^{2} \end{array}\right] \end{array}$$

The discrete measurement model is as follows:(17)$$\begin{array}{c} \underset{z[n]}{\underset{︸}{\left[\begin{array}{c} z_{\tau }\left[n\right]\\ z_{\phi }\left[n\right]\\ z_{f}\left[n\right] \end{array}\right]}}=\underset{H}{\underset{︸}{\left[\begin{array}{cccc} 1 & 0 & 0 & 0\\ 0 & 1 & 0 & 0\\ 0 & 0 & 1 & 0 \end{array}\right]}}A_{d}x\left[n-1\right]+\underset{v[n]}{\underset{︸}{\left[\begin{array}{c} v_{\tau }\\ v_{\phi }\\ v_{f} \end{array}\right]}} \end{array}$$

The innovation vector δz is represented as in Equation (Equation 8):(18)$$\begin{array}{c} \delta z\left[n\right]=z\left[n\right]-\hat{z}\left[n\right] \end{array}$$

The measurement covariance matrix R is the same as in Equation (Equation 5).

The system and measurement models in Equations (Equation 15) and (Equation 17) present four states, p=4, and three measurements, m=3. The open-loop discrete SSM is represented as:(19)x[n]=Adx[n−1]+ν3α3β3ν2α2β2ν1α1β1ν0α0β0τint︸Kdδτ[n]δϕ[n]δf[n]ν3α3β3ν2α2β2ν1α1β1ν0α0β0︸δz[n]
(20)$$\begin{array}{c} \underset{\hat{z}\left[n\right]}{\underset{︸}{\left[\begin{array}{c} {\hat{z}}_{\tau }\left[n\right]\\ {\hat{z}}_{\phi }\left[n\right]\\ {\hat{z}}_{f}\left[n\right] \end{array}\right]}}=H\;A_{d}\;x\left[n-1\right] \end{array}$$

To calculate the steady-state Kalman gains in discrete domain $K_{{\mathrm{ss}}_{d}}$, the solution of the DARE can be derived [44,45]:(21)$$\begin{array}{cc} P_{\mathrm{ss}}\; & =A_{d}\;P_{\mathrm{ss}}\;A_{d}^{T}+Q_{d}-A_{d}\;P_{\mathrm{ss}}\;H^{T}{\left(H\;P_{\mathrm{ss}}\;H^{T}+R\right)}^{-1}H\;P_{\mathrm{ss}}\;A_{d}^{T} \end{array}$$

Different methods such as the Schur decomposition can be used to solve the DARE [48]. This research takes a simplified approach using the analog Kalman gain coefficients $K_{\mathrm{ss}}$ derived from the CARE (see Equation (Equation 14)) in the discrete Kalman gain $K_{{\mathrm{ss}}_{d}}$:(22)$$\begin{array}{cc} K_{{\mathrm{ss}}_{d}} & \approx \tau_{\mathrm{int}}K_{\mathrm{ss}}=\tau_{\mathrm{int}}\left[\begin{array}{ccc} \kappa & \upsilon 2\gamma & \upsilon 2\gamma^{2}\frac{R_{\phi }}{R_{f}}\\ 0 & 2\gamma & 2\gamma^{2}\frac{R_{\phi }}{R_{f}}\\ 0 & 2\gamma^{2} & 3\gamma^{3}\frac{R_{\phi }}{R_{f}}\\ 0 & \gamma^{3} & 2\gamma^{4}\frac{R_{\phi }}{R_{f}} \end{array}\right] \end{array}$$

Figure 2 shows the linear model of the discrete full LUT-DSKF tracking architecture, which consists of three main components: the comparator, the loop filters (i.e., for DLL, FLL, and PLL), and the NCO. The comparator is the innovation stage of the DSKF (see Equation (Equation 18)), and the rest of the modules perform the state prediction and update of the DSKF (see Equations (Equation 19) and (Equation 20)). The steady-state Kalman gains are the loop filter coefficients (see Equation (Equation 22)). 

The open-loop transfer functions of the full LUT-DSKF are derived to show the match between the described system and the presented architecture. The code and carrier phase open-loop transfer functions results by combining the Z-transform of Equations (Equation 19) and (Equation 20):(23)$$\begin{array}{cc} \hat{z}\left(z\right) & =H\;A_{d}\;{\left(I-A_{d}\;z^{-1}\right)}^{-1}\;K_{{\mathrm{ss}}_{d}}\;z^{-1}\;\delta z\left(z\right) \end{array}$$

Equation (Equation 23) is derived as follows:(24)z^τ=τintz−11−z−1∑l=02τintl(1−z−1)l︸NCO(z)(νss3∑l=02τintl(1−z−1)l︸FDLL(z)δτ+υ∑l=02αss2−lτintl(1−z−1)l︸FPLL(z)δϕ+υ∑l=02βss2−lτintl(1−z−1)l︸FFLL(z)δf)
(25)$$\begin{array}{cc} {\hat{z}}_{\phi } & =\mathrm{NCO}\left(z\right)(F_{\mathrm{PLL}}\left(z\right)\delta \phi +F_{\mathrm{FLL}}\left(z\right)\delta f) \end{array}$$

In Equation (Equation 24), if PAD is disabled (i.e., υ=0), the FAP is uncoupled from the DLL. In this case, only the code phase difference δτ drives the DLL loop filter and the code NCO to output the estimated code phase measurement ${\hat{z}}_{\tau }$. On the contrary, if PAD is enabled (i.e., $\upsilon =f_{c}/f_{r}$), the carrier phase and carrier Doppler difference {δϕ,δf}, smoothed by the PLL and FLL loop filter $\left\{F_{\mathrm{PLL}},F_{\mathrm{PLL}}\right\}$, aids information to the code phase estimation.

Equation (Equation 25) shows the open-loop transfer function for the carrier phase estimation. The smoothed carrier Doppler, derived from the unsmoothed carrier phase and carrier Doppler error {δϕ,δf}, drives to the carrier NCO to obtain the estimated carrier phase ${\hat{z}}_{\phi }$. For simplicity, the linear model considers in the comparator the following relation between carrier Doppler and carrier phase:(26)$$\begin{array}{c} \delta f=\frac{1-z^{-1}}{\tau_{\mathrm{int}}}\delta \phi \end{array}$$

Two main findings can be addressed from the steady-state coefficients of the full LUT-DSKF. First, there is one single response time parameter γ for the FAP, significantly reducing the complexity of implementing an adaptive tracking technique. Second, the DLL has an additional time of response parameter κ. This parameter must be adapted independently, which implies an increase in complexity. In the following section, an extension of the LBCA adapts the time of response parameters of the full LUT-DSKF, {γ,κ}.

## 3. Adaptive Full Direct-State Kalman Filter

This section describes the architecture of the LBCA-based full LUT-DSKF. The LBCA updates the response time based on a weighted difference between estimated dynamics and noise statistics from the innovation vector [34]. In previous studies, the LBCA has been implemented in the standard STL [31,36], the DSKF [24], and the LUT-DSKF in a PLL [25] and a FAP [26] tracking scheme. Furthermore, this algorithm has been implemented in the interference mitigation stage to adapt the FLL of an adaptive notch filter (NF) [38,39,49].

The LBCA can update any parameter related to the system’s time of response. Equation (Equation 22) shows that γ and κ are related to the coefficients of $K_{{\mathrm{ss}}_{d}}$. Since there are two times of the response parameters, two LBCAs are required.

Figure 3 shows the architecture of the LBCA to adapt γ and κ. First, the normalized dynamics of the carrier phase error ${\bar{D}}_{\delta \phi }$ and the code phase error ${\bar{D}}_{\delta \tau }$ are calculated:(27)$${\bar{D}}_{\delta \phi }\left[n\right]=\frac{|\mu_{\delta \phi }\left[n\right]|}{|\mu_{\delta \phi }\left[n\right]|+\sigma_{\delta \phi }\left[n\right]}$$
(28)$${\bar{D}}_{\delta \tau }\left[n\right]=\frac{|\mu_{\delta \tau }\left[n\right]|}{|\mu_{\delta \tau }\left[n\right]|+\sigma_{\delta \tau }\left[n\right]}$$
where $\left\{|\mu_{\delta \phi }|,|\mu_{\delta \tau }|\right\}$ is the absolute mean and $\left\{\sigma_{\delta \phi },\sigma_{\delta \tau }\right\}$ the standard deviation of the carrier phase and code phase discriminator output, respectively. Second, the difference between the normalized dynamics and weighting function is performed:
(29)$$c_{\mathrm{FAP}}\left[n\right]=g_{{\mathrm{FAP}}_{\mathrm{Max}}}\;D_{\delta \phi }\left[n\right]-g_{\mathrm{FAP}}\left[n,\;\gamma \tau_{\mathrm{int}}\right]$$
(30)$$c_{\mathrm{DLL}}\left[n\right]=g_{{\mathrm{DLL}}_{\mathrm{Max}}}\;D_{\delta \tau }\left[n\right]-g_{\mathrm{DLL}}\left[n,\;\kappa \tau_{\mathrm{int}}\right]$$
where $\left\{c_{\mathrm{FAP}},c_{\mathrm{DLL}}\right\}$ are the control values use to update the {γ,κ} response times. $\left\{g_{\mathrm{FAP}},g_{\mathrm{DLL}}\right\}$ are the weighting functions that depend on the product between the integration time $\tau_{\mathrm{int}}$ and the response time parameter {γ,κ}. The maximum values of $\left\{g_{\mathrm{FAP}},g_{\mathrm{DLL}}\right\}$ are defined as $\left\{g_{{\mathrm{FAP}}_{\mathrm{Max}}},g_{{\mathrm{DLL}}_{\mathrm{Max}}}\right\}$. The control logic module accumulates the control values until there is an update of the response time parameter:(31)$$c_{\mathrm{FAP}}^{\mathrm{acc}}\left[n\right]=\left\{\begin{array}{cc} c_{\mathrm{FAP}}\left[n\right]+c_{\mathrm{FAP}}\left[n-1\right] & \mathrm{if}\;\gamma [n]=\gamma [n+1]\\ c_{\mathrm{FAP}}\left[n\right] & \mathrm{otherwise} \end{array}\right.$$
(32)$$c_{\mathrm{DLL}}^{\mathrm{acc}}\left[n\right]=\left\{\begin{array}{cc} c_{\mathrm{DLL}}\left[n\right]+c_{\mathrm{DLL}}\left[n-1\right] & \mathrm{if}\;\kappa [n]=\kappa [n+1]\\ c_{\mathrm{DLL}}\left[n\right] & \mathrm{otherwise} \end{array}\right.$$
where $\left\{c_{\mathrm{FAP}}^{\mathrm{acc}},c_{\mathrm{DLL}}^{\mathrm{acc}}\right\}$ are the accumulated control values. Finally, the accumulated control values update the current parameters {γ[n],κ[n]}:(33)$$\hat{\gamma }\left[n\right]=\gamma \left[n\right]+c_{\mathrm{FAP}}^{\mathrm{acc}}\left[n\right]$$
(34)$$\hat{\kappa }\left[n\right]=\kappa \left[n\right]+c_{\mathrm{DLL}}^{\mathrm{acc}}\left[n\right]$$
where $\left\{\hat{\gamma },\hat{\kappa }\right\}$ are the estimated response time parameters. A Schmitt trigger is used to avoid possible noise instabilities from $\left\{\hat{\gamma },\hat{\kappa }\right\}$:(35)$$\gamma \left[n+1\right]=\left\{\begin{array}{cc} \frac{6}{5}B_{{\mathrm{PLL}}_{0}} & \mathrm{if}\;n=0\\ \hat{\gamma }\left[n\right]+\Delta_{\mathrm{FAP}} & \mathrm{if}\;\hat{\gamma }\left[n\right]-\gamma \left[n\right]\geq \Delta_{\mathrm{FAP}}\\ \hat{\gamma }\left[n\right]-\Delta_{\mathrm{FAP}} & \mathrm{if}\;\gamma \left[n\right]-\hat{\gamma }\left[n\right]\leq \Delta_{\mathrm{FAP}}\;\\ \gamma [n] & \mathrm{otherwise} \end{array}\right.$$
(36)$$\kappa \left[n+1\right]=\left\{\begin{array}{cc} 4B_{{\mathrm{DLL}}_{0}} & \mathrm{if}\;n=0\\ \hat{\kappa }\left[n\right]+\Delta_{\mathrm{DLL}} & \mathrm{if}\;\hat{\kappa }\left[n\right]-\kappa \left[n\right]\geq \Delta_{\mathrm{DLL}}\\ \hat{\kappa }\left[n\right]-\Delta_{\mathrm{DLL}} & \mathrm{if}\;\kappa \left[n\right]-\hat{\kappa }\left[n\right]\leq \Delta_{\mathrm{DLL}}\;\\ \kappa [n] & \mathrm{otherwise} \end{array}\right.$$
where $\left\{\Delta_{\mathrm{FAP}},\Delta_{\mathrm{DLL}}\right\}$ are the update steps set to {0.5,0.01} Hz, and $\left\{B_{{\mathrm{PLL}}_{0}},B_{{\mathrm{DLL}}_{0}}\right\}$ are the initial loop bandwidths of the PLL and the DLL set to {8,1} Hz.

Moreover, the standard deviation estimation of the frequency discriminator output is required to calculate the ratio between $R_{\phi }$ and $R_{f}$. Due to this operation, the LBCA used in the FAP requires an extra division and multiplication compared to the LBCA implemented in the PLL [25].

Figure 4 presents the selected weighting function for the FAP, $g_{\mathrm{FAP}}$ and the DLL, $g_{\mathrm{DLL}}$:$$\begin{array}{cc} g_{\mathrm{FAP}}\left[\gamma \tau_{\mathrm{int}}\right] & =g_{{\mathrm{FAP}}_{\mathrm{Max}}}{\left[\begin{array}{c} T_{\mathrm{FAP}}\\ 1-T_{\mathrm{FAP}} \end{array}\right]}^{T}\left[\begin{array}{c} \mathrm{Sig}\left(50\left(\gamma \tau_{\mathrm{int}}-0.06\right)\right)\\ \mathrm{Sig}\left(250\left(\gamma \tau_{\mathrm{int}}-0.36\right)\right) \end{array}\right] \end{array}$$
(37)$$\begin{array}{c} =0.1{\left[\begin{array}{c} 0.14\\ 0.86 \end{array}\right]}^{T}\left[\begin{array}{c} \mathrm{Sig}\left(50\left(\gamma \tau_{\mathrm{int}}-0.06\right)\right)\\ \mathrm{Sig}\left(250\left(\gamma \tau_{\mathrm{int}}-0.36\right)\right) \end{array}\right] \end{array}$$
$$\begin{array}{cc} g_{\mathrm{DLL}}\left[\kappa \tau_{\mathrm{int}}\right] & =g_{{\mathrm{DLL}}_{\mathrm{Max}}}{\left[\begin{array}{c} T_{\mathrm{DLL}}\\ 1-T_{\mathrm{DLL}} \end{array}\right]}^{T}\left[\begin{array}{c} \mathrm{Sig}\left(200\left(\kappa \tau_{\mathrm{int}}-0.002\right)\right)\\ \mathrm{Sig}\left(250\left(\kappa \tau_{\mathrm{int}}-0.1\right)\right) \end{array}\right] \end{array}$$
(38)$$\begin{array}{c} =0.001{\left[\begin{array}{c} 0.4\\ 0.6 \end{array}\right]}^{T}\left[\begin{array}{c} \mathrm{Sig}\left(200\left(\kappa \tau_{\mathrm{int}}-0.002\right)\right)\\ \mathrm{Sig}\left(250\left(\kappa \tau_{\mathrm{int}}-0.1\right)\right) \end{array}\right] \end{array}$$
where Sig(·) is the Sigmoid function [50], and $\left\{T_{\mathrm{FAP}},T_{\mathrm{DLL}}\right\}$ are the dynamic thresholds of $\left\{g_{\mathrm{PLL}},g_{\mathrm{DLL}}\right\}$. The lower the dynamic threshold, the more sensitive the LBCA is to dynamics. On the contrary, a high dynamic threshold implies a higher sensitivity to noise. To reduce the Sigmoid function complexity, the piecewise linear approximation of nonlinearities (PLAN) technique is used [31,51] to piece-wise interpolate it. The weighting function depends on the normalized bandwidth $B_{N}$, which represents the product between the loop bandwidth and the integration time $\left\{\gamma \tau_{\mathrm{int}},\kappa \tau_{\mathrm{int}}\right\}$.

Figure 5 presents the architecture of the LBCA-based full LUT-DSKF. Compared to Figure 2, only the ${\mathrm{LBCA}}_{\mathrm{DLL}}$ and the ${\mathrm{LBCA}}_{\mathrm{FAP}}$ are added to adapt the steady state coefficients of the full LUT-DSKF (see Equation (Equation 22)).

## 4. Experimental Setup

This section describes the GNSS receiver under test, the metric used to determine the system performance, and the simulated scenarios.

### 4.1. GNSS Receiver

The GOOSE© platform, developed by Fraunhofer IIS and marketed through TeleOrbit GmbH, is a GNSS receiver with an open software interface [13,52,53]. This receiver contains a customized tri-band radio-frequency front-end (RFFE), a Xilinx Kintex7 field-programmable gate array (FPGA), and a peripheral component interconnect express (PCIe) interface to connect to an external processor. Figure 6 shows the GOOSE single board computer (SBC) receiver. The RFFE amplifies, filters, downconverts, discretizes the GNSS signals, and sends the digital samples to the FPGA. The analog-to-digital converter (ADC) discretizes each frequency band at a sample rate of 81 MHz and a resolution of 8 bits for the in-phase and quadrature-phase (IQ) components. The FPGA includes one acquisition module and sixty single point correlator (SPC) tracking channels, which the processor can control. The Kintex7 FPGA of the GOOSE SBC is connected to a dual-core ARM processor with an Ubuntu 16.04 operating system and 1GB RAM. The processor performs the acquisition of the incoming digital samples using the acquisition module of the FPGA. The tracking starts once the acquisition achieves a rough estimate of the frequency Doppler *f* and code phase τ. The tracking stage of this GNSS receiver is partially implemented in the FPGA (e.g., correlators and NCO) and software (e.g., discriminators and loop filters). This stage consists of three steps. First, the FLL and the DLL refine the acquired *f* and τ estimates. Second, the PLL starts and synchronizes with the carrier phase. Finally, the FLL stops and the PLL can work unaided when the latter successfully achieves a good lock with the carrier phase. The receiver synchronizes with the navigation data at this stage, and the integration time increases to the symbol period. In the case of Global Positioning System (GPS) L1 C/A, the integration time is increased to 20 ms. Table 1 presents the configuration of the tracking scheme during the fine carrier phase synchronization. Once the navigation data is decoded, the PVT solution is computed based on the pseudo range measurements.

The LBCA-based full LUT-DSKF is implemented in the tracking stage of the GOOSE receiver in software. In order to evaluate correctly the performance of this tracking architecture, the reacquisition is disabled.

### 4.2. System Performance Metric

Two metrics are selected to evaluate the system performance of the proposed adaptive tracking architecture. The first metric evaluates the system performance in terms of the carrier phase. The carrier system performance $P_{\phi }$ is the same as in previous studies [25]:(39)$$\begin{array}{c} P_{\phi }=\bar{\mathrm{PLI}}\times {\bar{N}}_{\mathrm{sat}} \end{array}$$
where $\bar{\mathrm{PLI}}$ denotes the phase-lock indicator (PLI) average between the satellites that remain on track, and ${\bar{N}}_{\mathrm{sat}}$ indicates the normalized minimum number of tracked visible satellites during the entire simulation.

The expression of $\bar{\mathrm{PLI}}$ is:(40)$$\bar{\mathrm{PLI}}=\frac{1}{K_{\mathrm{sim}}\;N_{\mathrm{sat}}^{\min}}\sum_{k=k_{0}}^{k_{0}+K_{\mathrm{sim}}}\sum_{l=1}^{N_{\mathrm{sat}}^{\min}}{\mathrm{PLI}}^{l}\left[k\right]$$
where $K_{\mathrm{sim}}$ is the number of measurement epochs under evaluation, and $k_{0}$ is the starting time in samples. $N_{\mathrm{sat}}^{\min}$ is the minimum number of visible satellites that remain on track during the simulation time under evaluation:(41)$$N_{\mathrm{sat}}^{\min}=\min\left(N_{\mathrm{sat}}\left[k\right]\right)\;\forall k=k_{0},\cdots ,k_{0}+K_{\mathrm{sim}}$$

The ${\mathrm{PLI}}^{l}\left[k\right]$ of the *l*th SV being tracked is calculated based on the prompt IQ components $\left\{I_{p}^{l},Q_{p}^{l}\right\}$ [25]:(42)$${\mathrm{PLI}}^{l}\left[k\right]=\frac{{\left(I_{p}^{l}\left[k\right]\right)}^{2}-{\left(Q_{p}^{l}\left[k\right]\right)}^{2}}{{\left(I_{p}^{l}\left[k\right]\right)}^{2}+{\left(Q_{p}^{l}\left[k\right]\right)}^{2}}$$

The second term of Equation (Equation 39), ${\bar{N}}_{\mathrm{sat}}$, is defined as:(43)$$\begin{array}{c} {\bar{N}}_{\mathrm{sat}}=\frac{N_{\mathrm{sat}}^{\min}}{N_{\mathrm{sat}}^{\mathrm{total}}} \end{array}$$
where $N_{\mathrm{sat}}^{\mathrm{total}}$ is the total number of visible SVs during the simulation.

The second metric evaluates the system performance in terms of the code phase. The GOOSE SBC has been configured only to compute the PVT based on the pseudo ranges derived from the code phase estimates. The horizontal root mean square error (HRMSE) gives a good indicator of the code system performance. To avoid infinite values of the HRMSE in case there is no fix of the PVT solution during the simulation time under evaluation, the inverse of the HRMSE is considered. Then, the code system performance $P_{\tau }$ is defined as:(44)$$\begin{array}{c} P_{\tau }=\frac{1m}{\mathrm{HRMSE}}=E{\left[\sqrt{{\left(r^{N}-r_{G}^{N}\right)}^{2}+{\left(r^{E}-r_{G}^{E}\right)}^{2}}\right]}^{-1} \end{array}$$
where E[·] is defined as the mean operation, $r^{N}$ and $r^{E}$ are the calculated north and east user position from the GOOSE platform, and $r_{G}^{N}$ and $r_{G}^{E}$ are the north and east ground truth user position.

$P_{\phi }$ and $P_{\tau }$ are both unitless. A high value of $P_{\phi }$ and $P_{\tau }$ indicates a good system performance. The opposite case means an increased probability of losing the PVT fix. A final metric is achieved combining (Equation 39) and (Equation 44):(45)$$\begin{array}{c} P_{\mathrm{system}}=P_{\phi }\cdot P_{\tau } \end{array}$$

The average system performance ${\bar{P}}_{\mathrm{system}}$ with respect the *C*/*N*_0_ levels determines the overall performance of the adaptive tracking technique.
(46)$$\begin{array}{c} {\bar{P}}_{\mathrm{system}}=\sum_{k=1}^{N_{\mathrm{CN}0}}\frac{P_{\mathrm{system}}^{k}}{N_{\mathrm{CN}0}} \end{array}$$
where $N_{\mathrm{CN}0}$ is the number of *C*/*N*_0_ levels. The system performance metric ${\bar{P}}_{\mathrm{System}}$ which accounts for both noise and dynamics for tracking, is a novel contribution of this paper.

### 4.3. Evaluation Setup

Figure 7 shows a block diagram of the evaluation setup. It is the same as in previous studies [25,31,34,37]. The setup consists of three main parts: the Spirent GSS9000 radio frequency constellation simulator (RFCS), the GOOSE SBC, and the user computer. The user computer manages the simulator and the GOOSE SBC through transmission control protocol (TCP) to perform the test automation. First, the user computer configures RFCS. It selects the desired scenario and sets the *C*/*N*_0_ to 50 dBHz to all the GNSS signals. A high *C*/*N*_0_ level is selected to ensure that all the visible SVs achieve tracking at the beginning of the scenario. Second, once the RFCS is ready and starts the simulation ($T_{\mathrm{sim}}=0\;s$), the user sends the application to the GOOSE SBC and commands the GOOSE to run it. Third, the user reduces the *C*/*N*_0_ level by 4 dB until reaching the desired *C*/*N*_0_ level. Finally, after 20 min of simulation ($T_{\mathrm{sim}}=1200\;s$), the user stops the application that is running in GOOSE, collects all the logging data, and stops the simulation at the RFCS. The process repeats until evaluating all the desired *C*/*N*_0_ levels for all the applications, and for all the selected scenarios.

The selected *C*/*N*_0_ levels are {26, 30, 34, 38, 42, 46, 50} dBHz. The first 10 min of the simulations, $T_{\mathrm{sim}}=\left\{0-600\right\}\;s$, are used to reach the desired *C*/*N*_0_ level and, in case of the adaptive tracking, to reach also to stability in the response time. The last 10 min of the simulation, $T_{\mathrm{sim}}=\left\{600-1200\right\}\;s$, are under evaluation. Considering that the sampling period of the logged measurements is equal to the integration time $\tau_{\mathrm{int}}$, 20 ms, the starting evaluation time $k_{0}$ and the simulation time under evaluation $K_{\mathrm{sim}}$ are:(47)$$k_{0}=\frac{600\;s}{\tau_{\mathrm{int}}\times 10^{-3}}=30,000\;\mathrm{samples}$$
(48)$$K_{\mathrm{sim}}=\frac{1200\;s}{\tau_{\mathrm{int}}\times 10^{-3}}-k_{0}=30,000\;\mathrm{samples}$$

A static scenario and a dynamic scenario are selected to evaluate these applications. In both scenarios, the radiation pattern of the antenna is simulated as isotropic, and it is direct line-of-sight (LOS) with the SVs. In future work, different antenna patterns and more challenging environments will be simulated. Figure 8 shows the skyplot of both scenarios. There are 10 visible satellites during the simulation. However, SV G1 disappears behind the horizon after two minutes of simulation, SV G30 rises above the horizon near the end of the simulation, and SV G23 is discarded due to its low elevation. Therefore, the maximum number of visible satellites $N_{\mathrm{sat}}^{\mathrm{total}}$ is limited to seven.

Figure 9 presents the LOS dynamics of the simulated dynamic scenario. During the first 10 min of the simulation, the vehicle remains static. During the second half of the simulation, the vehicle moves at high speed generating some jerk dynamics that can challenge the tracking stage. In this scenario, the maximum LOS signal jerk dynamics is 11.55 g/s in SV G9.

From Figure 5, different configurations of this adaptive tracking architecture can be considered. For instance, the ${\mathrm{LBCA}}_{\mathrm{FAP}}$ and the ${\mathrm{LBCA}}_{\mathrm{PAD}}$ can be bypassed. Furthermore, the FAP can be disabled by setting the FLL coefficients to zero once the PLL achieves lock. Furthermore, the PAD can be enabled or disabled based on Equation (Equation 2). Therefore, in this research 10 applications with different configurations are selected to be evaluated. Table 2 shows the different tracking schemes derived from the LBCA-based LUT-DSKF. These applications are evaluated under different dynamics and noise levels. Since there are seven *C*/*N*_0_ levels and two scenarios, the total amount of time required to evaluate a single application is 280 min.

A theoretical method to quantify an adaptive tracking technique’s complexity is to measure the number of required mathematical operations. This method provides a “best-case” comparison and neglects any implementation limitations. Table 3 presents the theoretical complexity based on the added number of additions, multiplications, and divisions for each tracking configuration. The ${\mathrm{LBCA}}_{\mathrm{FAP}}$ includes an additional multiplication and division compared to the ${\mathrm{LBCA}}_{\mathrm{DLL}}$ due to the ratio $R_{\phi }/R_{f}$ used to adapt the FLL coefficients (see Equation (Equation 22). The aiding the FLL into the PLL (i.e., FAP) adds three additions and three multiplications (see Figure 2). The aiding of the estimated carrier Doppler to the DLL (i.e., PAD) adds only one addition. The division presented in Equation (Equation 2) can be precomputed during initialization.

## 5. Results

The results are separated into three sections. First, the system performance of the static scenario is evaluated. Second, the system performance of the dynamic scenario is presented. Finally, the average system performance of each adaptive tracking configuration is summarized in a table. The dataset used to plot the presented results is available online to download [54].

### 5.1. Static Scenario

Figure 10 shows the carrier system performance $P_{\phi }$ and the code system performance $P_{\tau }$ in a static scenario under different *C*/*N*_0_ levels. The selected tracking configurations under evaluation are listed in Table 2. All the tracking configurations present similar carrier and code system performance. The only tracking configuration that loses the PVT solution is the LBCA-based PLL with an unaided LBCA-based DLL. The LBCA’s weighting function used for the carrier phase synchronization (see Equation (Equation 37)) is configured to be sensitive to dynamics. Therefore, low *C*/*N*_0_ levels can challenge carrier synchronization with this configuration. However, this is a sporadic error since the other tracking configurations using the same LBCA can maintain a PVT fix.

An interesting result can be observed in Figure 10b. When no PAD is enabled, the LBCA-based DLL performs poorly compared to the other tracking techniques at any *C*/*N*_0_ level. The LBCA used for the DLL is highly noise-sensitive, leading to a constant decrease in the DLL bandwidth until reaching a minimum bandwidth of 0.25 Hz. Since there is no carrier aiding to mitigate the dynamics, the DLL suffers a code phase bias error. Although being a static scenario, these dynamics can be generated by the receiver’s clock. For more extended simulations, this tracking configuration probably loses the PVT fix.

### 5.2. Dynamic Scenario

Figure 11 depicts the dynamic system performance of the selected tracking configurations. The standard tracking techniques have no PVT fix at any *C*/*N*_0_ level. The LBCA-based PLL techniques do not maintain either the PVT fix, but they manage to keep the tracking of at least two to three SVs from 34 dBHz to 50 dBHz. The high $P_{\phi }$ score compared to the standard tracking techniques can be observed in Figure 11a. Only the LBCA-based FAP architectures keep a continuous PVT solution from 34 dBHz to 50 dBHz. From the LBCA-based FAP techniques, the one with the standard DLL presents the best carrier system performance. At 30 dBHz, although the PVT fix is lost, this tracking technique maintains a continuous track of three SVs during the entire simulation. The LBCA-based FAP combined with the LBCA-based DLL presents a continuous PVT from 34 dBHz to 50 dBHz, but Figure 11b shows a degraded $P_{\tau }$ compared to the rest of LBCA-based FAP architectures. The reason behind this is the low bandwidth of the DLL set by theLBCA and the unaided carrier dynamics.

### 5.3. Total System Performance

Table 4 summarizes the final static and dynamic system performance of each tracking configuration under evaluation. The best static and dynamic system performance is marked green, whereas the worse performance is marked red. Moreover, the added time complexity of each adaptive tracking technique is included. The same procedure to calculate the added time complexity as in previous research is carried out [25,31]. The complexity is marked as red, orange, and green, depending on the level of complexity. The colors vary from the most complex one, marked in red, to the simplest one, marked in green.

The main outcome of these results is the significant improvement of the system performance in the high dynamic scenario using the LBCA-based FAP. It presents an excellent system performance in dynamic scenarios, keeping a great tracking sensitivity in static scenarios. This tracking configuration is the only one that maintains a continuous PVT solution during the entire high-dynamic scenario. The LBCA-based PLL and the standard techniques are not robust enough to maintain a continuous PVT solution. It is possible to improve the LBCA-based PLL to be more sensitive to dynamics by decreasing the dynamic threshold $T_{\mathrm{FAP}}$ of the LBCA weighting function (see Equation (Equation 37)). However, that change can degrade the tracking sensitivity in stationary scenarios. It highlights the trade-off for tuning for sensitivity to dynamic scenarios.

The system performance using the LBCA-based DLL could be improved. The high sensitivity to noise drives the selected time of response parameter κ to a low value. Some further tuning of its weighting function $g_{\mathrm{DLL}}$ is required. When carrier aiding is enabled, the LBCA-based FAP aids the dynamics into the DLL, achieving good scores in the system performance for both scenarios. However, among the LBCA-based FAP tracking schemes, the LBCA-based FAP with unaided DLL achieves the best performance. In the static scenario, aiding the carrier Doppler into the code phase estimation can be counterproductive at low *C*/*N*_0_ levels, since the carrier Doppler estimate becomes noisy. On the contrary, at high *C*/*N*_0_ levels, carrier aiding is a solution to decrease the DLL bandwidth and improve the code system performance. In the dynamic scenario, the carrier aiding was expected to outperform the other configurations. However, a slight decrease in performance is observed. Further testing involves fine-tuning the weighting function for the LBCA-based DLL.

A separate LBCA for the DLL increases the complexity. Further investigations will be conducted on reducing the adaptive full LUT-DSKF using one single LBCA. In addition, a negligible increase in complexity is observed while enabling PAD. The use of FAP also shows a minor increase.

The configuration with the best system performance is the LBCA-based FAP with unaided DLL, being 1.9 times more complex than the standard tracking, the second least complex from the presented techniques. Figure 12b and Figure 13 present the position estimation and the variation of the FAP’s response time parameter γ in the simulated static and dynamic scenario at 34 dBHz. Further analysis of the other configurations is available on the cloud [54]. Figure 12a shows the estimated position in the static scenario using this adaptive tracking configuration at 34 dBHz. Furthermore, the response time parameter γ is depicted in Figure 12b. Initially, the *C*/*N*_0_ level is 50 dBHz, and each LBCA increases γ to 13 Hz. Every 30 s, the *C*/*N*_0_ decreases until reaching 34 dBHz. The LBCA reduces the response time parameter accordingly. At 34 dBHz, the LBCA maintains a γ value of 8 Hz.

In the dynamic scenario, the comparison between the estimated trajectory based on this adaptive tracking configuration and the reference trajectory is shown in Figure 13a. The circles depicted in red indicate the high dynamic events with significant LOS jerk dynamics. Figure 13b presents the γ adaptation done by the LBCA. During the static region, each tracking channel’s LBCA lstabilizes the bandwidth to 8 Hz. Once dynamic events are present, marked in red circles, the LBCA increases the bandwidth to maintain the carrier phase lock. The bandwidth peaks are closely related to the LOS jerk dynamics of the simulated scenario presented in Figure 9b. After the dynamic events, the LBCA returns to a low bandwidth to maintain a good carrier phase synchronization at 34 dBHz.

These results confirm the robustness of the presented adaptive full LUT-DSKF to maintain carrier-phase continuity under different noise and signal dynamic levels while keeping low time complexity.

## 6. Conclusions

This paper presents a complete single-frequency adaptive scalar tracking architecture: the LBCA-based full LUT-DSKF. First, the full DSKF is analyzed by explaining the system and measurement model, the state space model, and the transfer function. Second, to reduce the complexity of the full DSKF, the convergence of the Kalman gains is calculated by solving the CARE, deriving the so-called full LUT-DSKF. Previous research shows that the LUT-DSKF reduces the time complexity by more than half compared to the DSKF [25]. This simplification shows a relationship between Kalman gains based on the ratio parameter γ and κ (see Equation (Equation 22)). Third, the response time of the full LUT-DSKF is adapted through γ and κ using two LBCAs. Fourth, the carrier and code system performance of different configurations of the LBCA-based full LUT-DSKF are compared in a static and a dynamic scenario under different *C*/*N*_0_ levels. A metric to evaluate the code system performance is presented based on the HRMSE. The product of the code and carrier system performance leads to a final metric in which the tracking scheme can be nicely evaluated. The results show the importance of the LBCA-based FAP in high dynamic scenarios. This tracking configuration is the only one that maintains the PVT solution, requiring a slightly increased complexity compared to the LBCA-based PLL.

The inter-dependency between FLL and PLL coefficients in a FAP architecture benefits the implementation of a low-complexity adaptive technique using a single LBCA. Another important observation is the fact that it is not necessary to set the third coefficient of the FLL, $\beta_{2}$, to zero, as it is usually done. While deriving the DARE, one can observe that $\beta_{2}$ equals zero is not the optimal configuration. No relationship between the DLL and the FAP coefficients has been found. Therefore, another LBCA is required to adapt the DLL coefficients.

Future work follows-up recent research [36] testing the presented adaptive tracking architecture in simulated rocket scenarios. Next studies consist of analyzing the effect of the sounding rocket’s attitude and its antenna’s radiation pattern in the adaptive full LUT-DSKF. Moreover, an extension of the presented tracking architecture is proposed: an LBCA-based multi-frequency adaptive tracking architecture. Multi-frequency tracking architectures imply a selective frequency diversity that can benefit tracking sensitivity [55]. The addition of the LBCA in this tracking architecture can optimize the tracking performance by weighting the filter coefficients depending on the estimated dynamics and noise of each band. Furthermore, as multipath affects each frequency band differently, multi-frequency adaptive tracking techniques can suppress the bands affected by multipath while allowing only the non-affected ones. The use of the LBCA in this multi-frequency tracking scheme cannot only improve the tracking performance in dynamic and noisy scenarios, but it can also increase the robustness against interferences and multipath effects.

Like the loop bandwidth, the integration time also affects the tracking response time. A method to adapt the integration time based on the LBCA’s loop-bandwidth update has been presented [41]. Future research will introduce the normalized-bandwidth control algorithm (NBCA): an extension of the LBCA that adapts the loop bandwidth and the integration time simultaneously. An improvement in tracking sensitivity is expected, particularly in pilot signals.

## Figures and Tables

**Figure 1 sensors-23-03658-f001:**
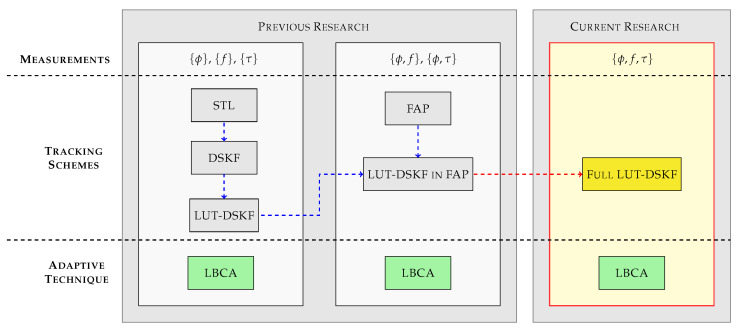
Research survey and comparison to other publications.

**Figure 2 sensors-23-03658-f002:**
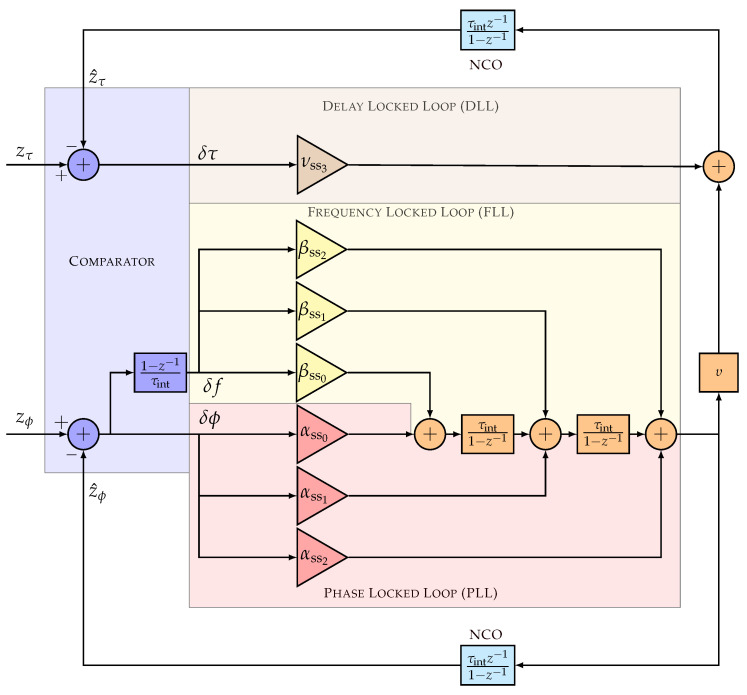
Linear model of full LUT-DSKF tracking architecture.

**Figure 3 sensors-23-03658-f003:**
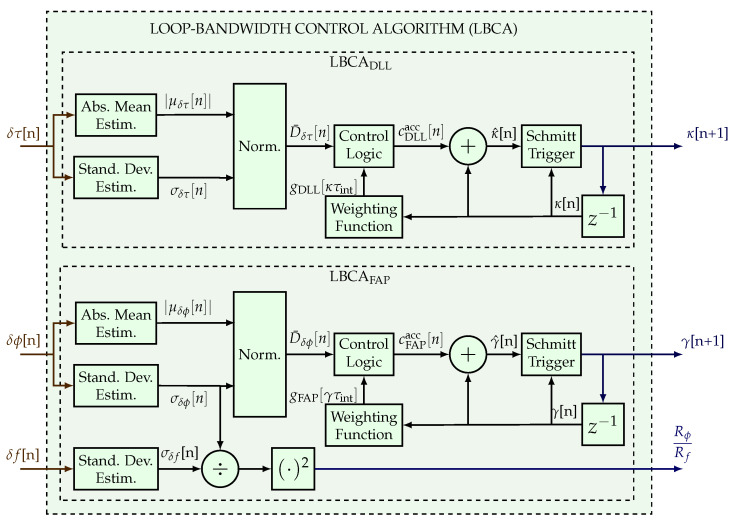
LBCA architecture used in the full LUT-DSKF.

**Figure 4 sensors-23-03658-f004:**
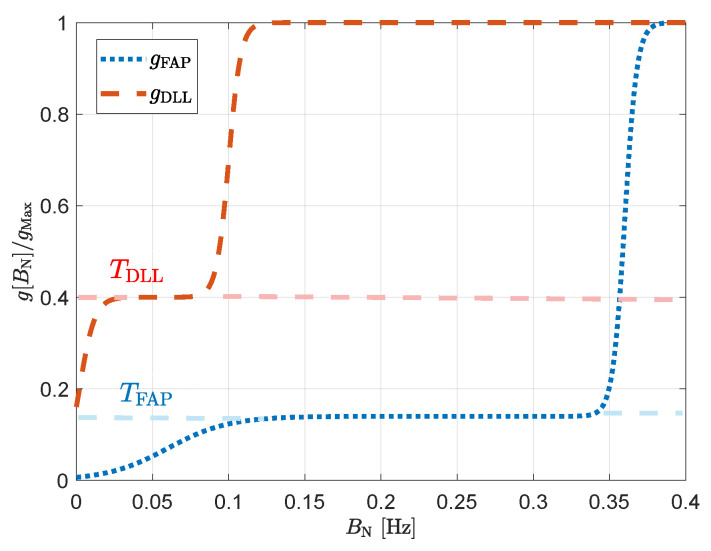
Selected normalized weighting functions for ${\mathrm{LBCA}}_{\mathrm{FAP}}$ and ${\mathrm{LBCA}}_{\mathrm{DLL}}$.

**Figure 5 sensors-23-03658-f005:**
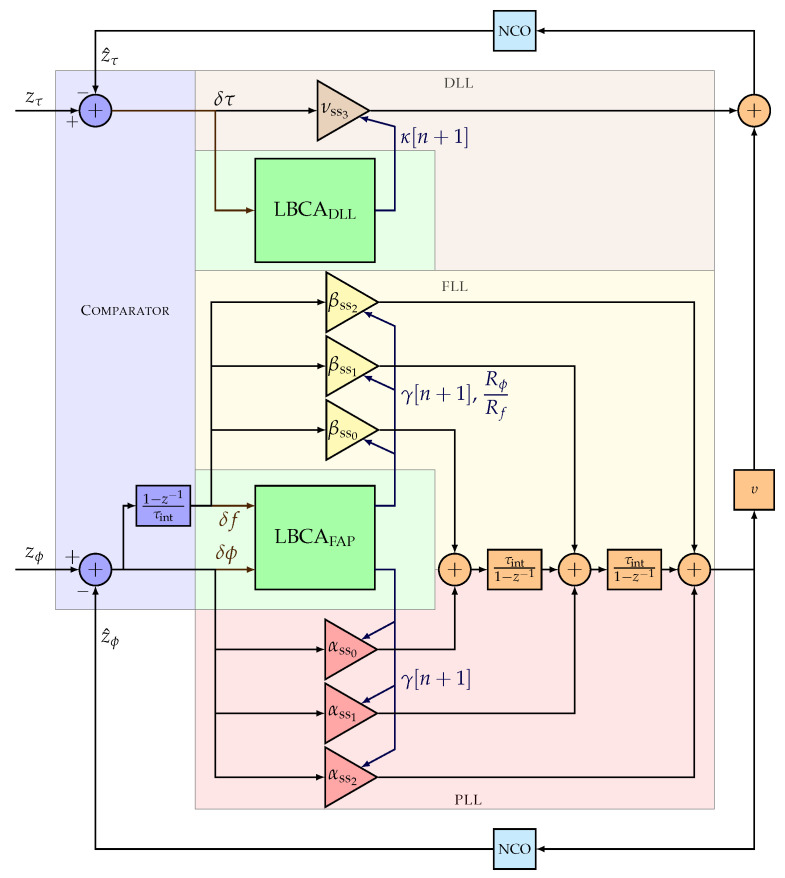
Adaptive full LUT-DSKF using LBCA.

**Figure 6 sensors-23-03658-f006:**
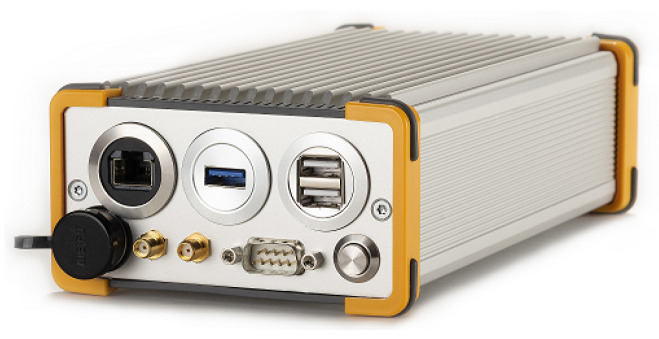
Photo of the GOOSE SBC receiver @Fraunhofer IIS/Paul Pulkert.

**Figure 7 sensors-23-03658-f007:**
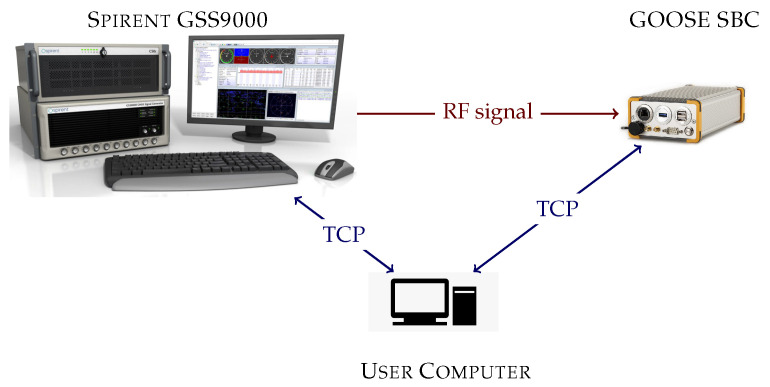
Evaluation setup consisting of a Spirent RFCS, a GOOSE SBC receiver, and a control computer.

**Figure 8 sensors-23-03658-f008:**
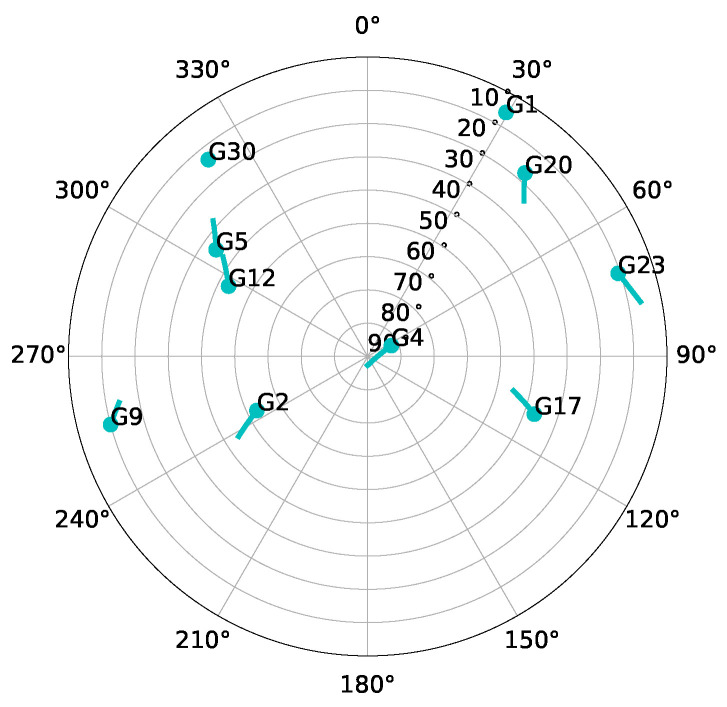
Sky-plot of the simulated scenarios.

**Figure 9 sensors-23-03658-f009:**
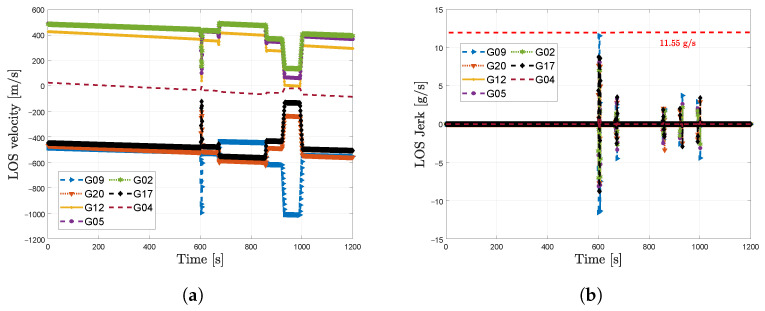
LOS dynamics in simulated dynamic scenario. (**a**) LOS velocity dynamics. (**b**) LOS jerk dynamics.

**Figure 10 sensors-23-03658-f010:**
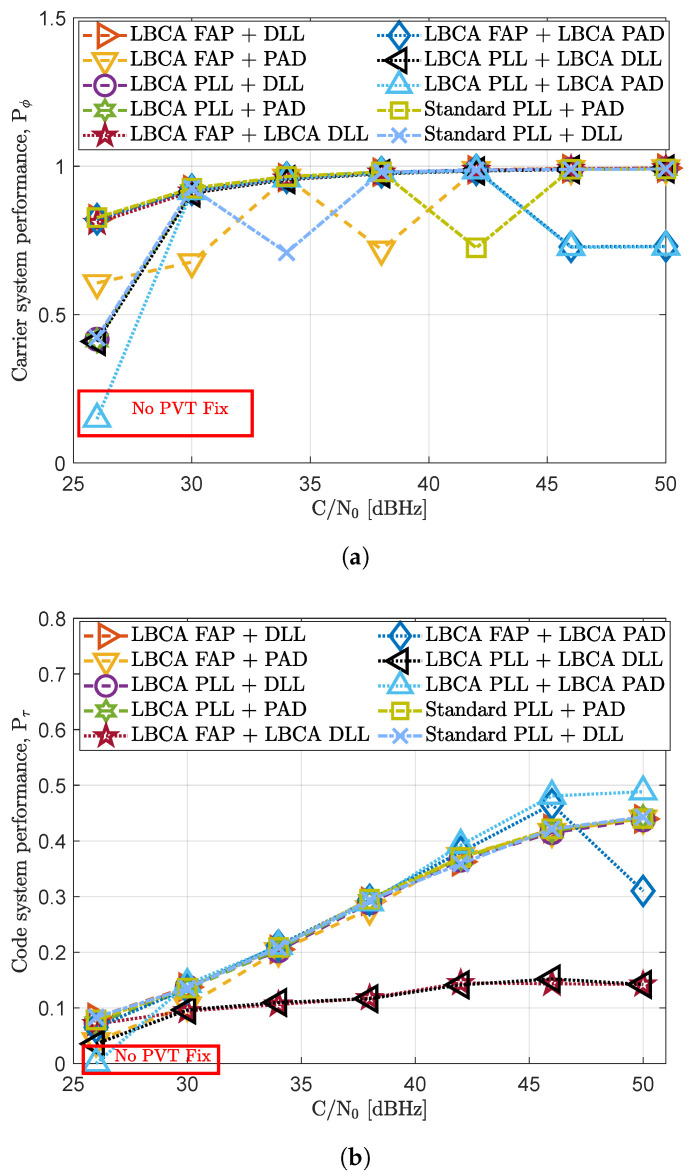
System performance evaluation in static scenario. (**a**) Carrier system performance. (**b**) Code system performance.

**Figure 11 sensors-23-03658-f011:**
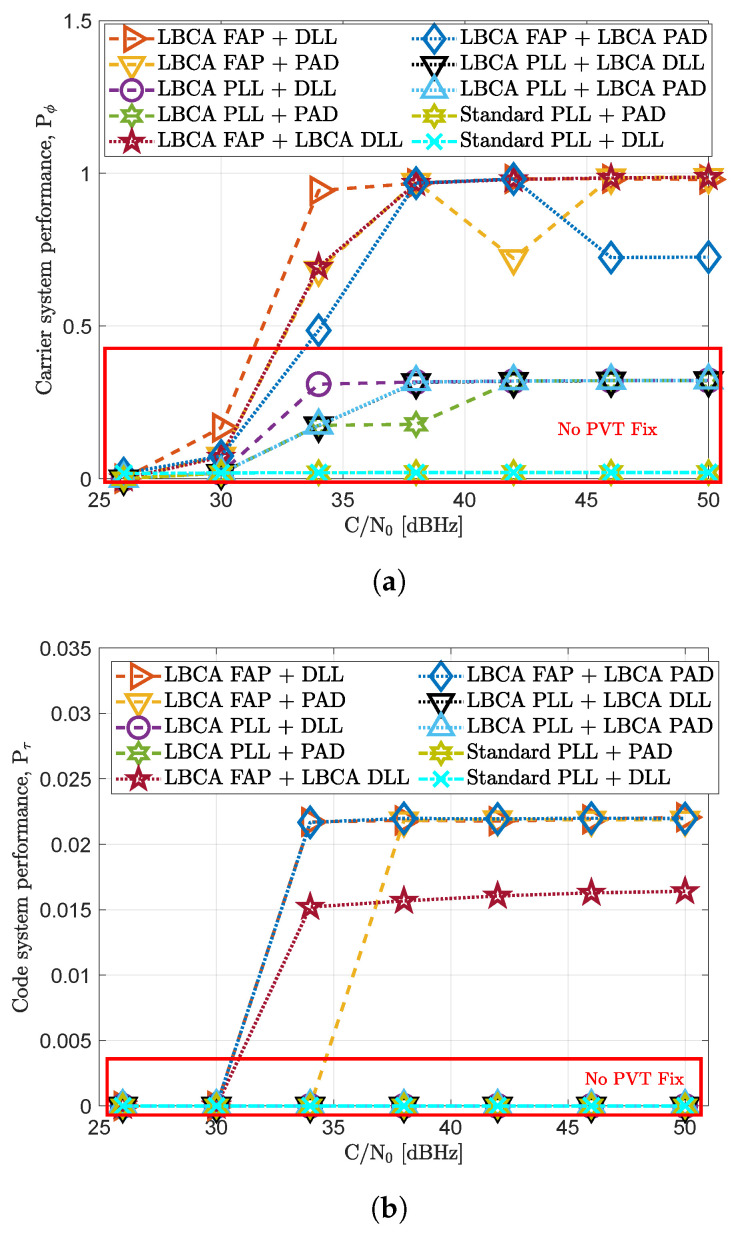
System performance evaluation in dynamic scenario. (**a**) Carrier system performance. (**b**) Code system performance.

**Figure 12 sensors-23-03658-f012:**
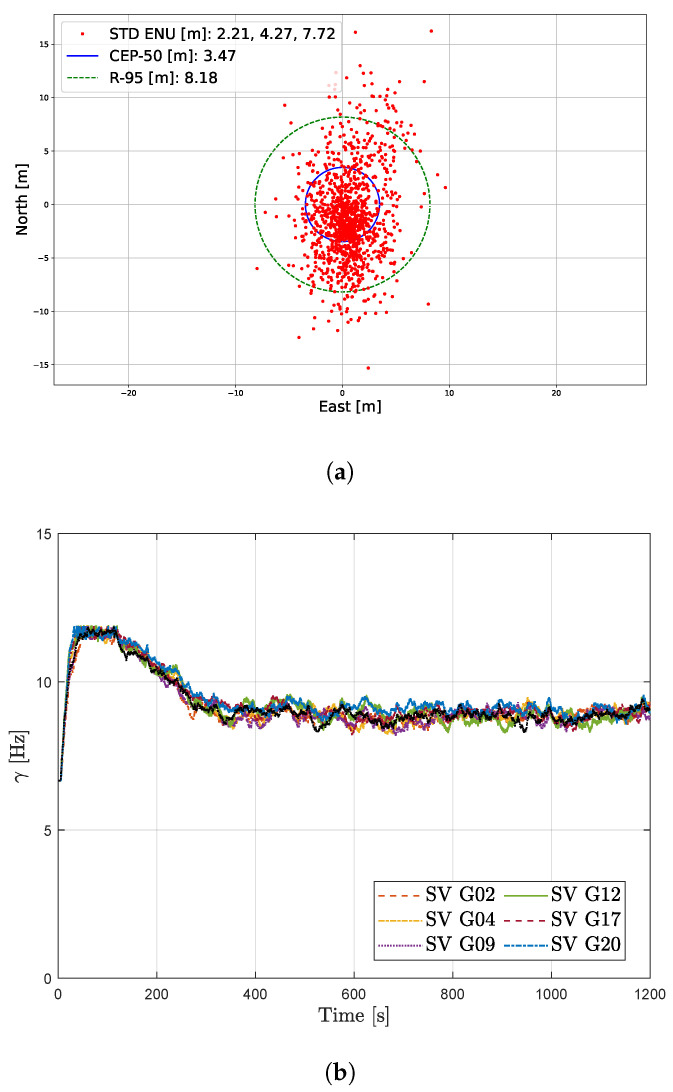
Results of LBCA-based FAP with unaided DLL at 34 dBHz in the static scenario. (**a**) Position point-cloud. (**b**) Loop-bandwidth variation of FAP architecture.

**Figure 13 sensors-23-03658-f013:**
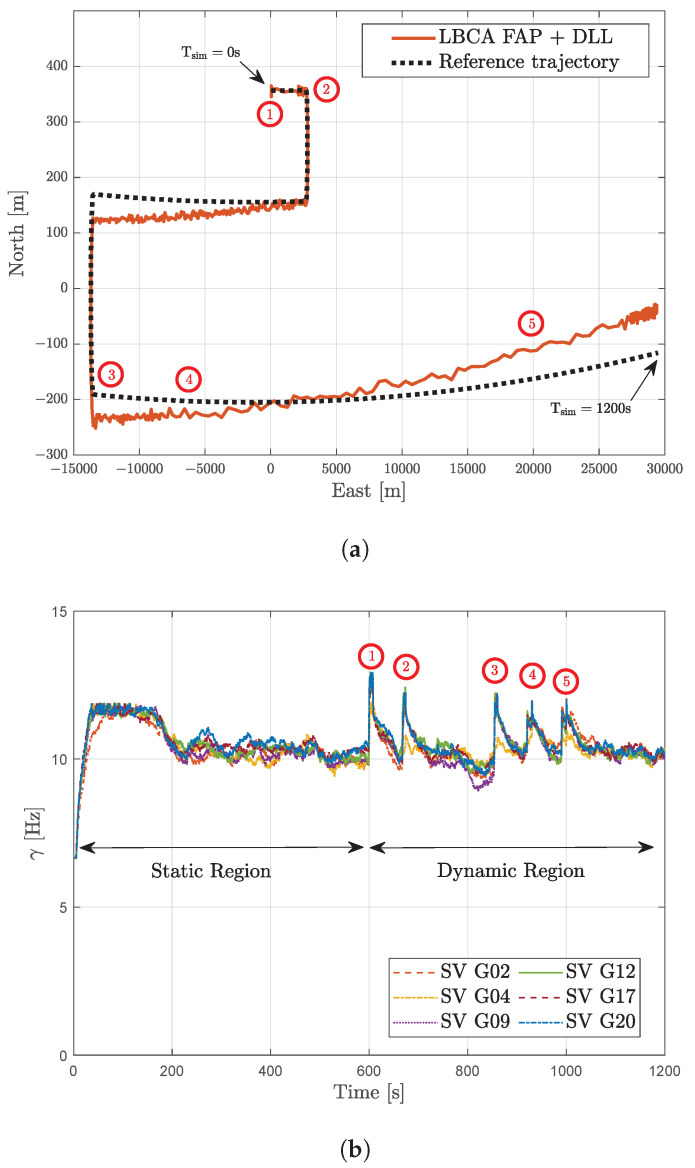
Results of LBCA-based FAP with unaided DLL at 34 dBHz in a dynamic scenario. (**a**) Comparison between reference trajectory and estimated trajectory from GOOSE. (**b**) Loop-bandwidth variation of FAP architecture.

**Table 1 sensors-23-03658-t001:** Configuration of tracking stage in GOOSE receiver.

Configuration Parameter	FLL	PLL	DLL
Discriminator type	Atan2(·)	Atan(·)	Dot product
Initial bandwidth [Hz]	0	8	1
Chip spacing, $\Delta_{s}$ [chips]		0.5	
Integration time, $\tau_{\mathrm{int}}$ [ms]		20	
GNSS signal		GPS L1 C/A	

**Table 2 sensors-23-03658-t002:** Tracking applications under evaluation. The tracking configurations used for each application are marked with x.

Tracking Scheme	Tracking Configuration:			
	${\mathrm{LBCA}}_{\mathrm{FAP}}$	${\mathrm{LBCA}}_{\mathrm{DLL}}$	FAP	PAD
LBCAFAP+DLL	x		x	
LBCAFAP+PAD	x		x	x
LBCAPLL+DLL	x			x
LBCAPLL+PAD	x			x
LBCAFAP+LBCADLL	x	x	x	
LBCAFAP+LBCAPAD	x	x	x	x
LBCAPLL+LBCADLL	x	x		x
LBCAPLL+LBCAPAD	x	x		x
StandardPLL+PAD				x
StandardPLL+DLL				

**Table 3 sensors-23-03658-t003:** Complexity of tracking configurations based on the added number of operations.

Tracking Configuration	Added Number of Operations:		
	Additions	Multiplications	Divisions
${\mathrm{LBCA}}_{\mathrm{FAP}}$ [31]	6	8	2
${\mathrm{LBCA}}_{\mathrm{DLL}}$ [31]	6	7	1
FAP	3	3	0
PAD	1	0	0

**Table 4 sensors-23-03658-t004:** System performance of adaptive tracking techniques.

Tracking	Static	Dynamic	Added Time
Technique	${\bar{P}}_{\mathrm{System}}$	${\bar{P}}_{\mathrm{System}}$	Complexity
LBCAFAP+DLL	0.0386 ${}^{*}$	0.0022	1.94 ${}^{‡}$
LBCAFAP+PAD	0.0349	0.0016	1.94
LBCAPLL+DLL	0.0375	0	1.90
LBCAPLL+PAD	0.0377	0	1.90
LBCAFAP+LBCADLL	0.0160	0.0015	2.84
LBCAFAP+LBCAPAD	0.0329	0.0017	2.84
LBCAPLL+LBCADLL	0.0153 ${}^{†}$	0	2.81
LBCAPLL+LBCAPAD	0.0348	0	2.81
StandardPLL+PAD	0.0368	0	1.00
StandardPLL+DLL	0.0369	0	1.00

Added time complexity is the factor that the algorithm takes to process compared to a standard tracking architecture. ${}^{*}$ Values in green indicate best performance or least added complexity. ${}^{†}$ Values in red indicate worst performance or most added complexity. ${}^{‡}$ Values in orange indicate medium added complexity.

## Data Availability

Publicly available datasets were analyzed in this study. This data can be found here: https://owncloud.fraunhofer.de/index.php/s/LGoWPVtV5xbQ9mB (accessed on 18 March 2023).

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
