# Peer review of "Evaluation of Low-Complexity Adaptive Full Direct-State Kalman Filter for Robust GNSS Tracking [Author-notes fn1-sensors-23-03658]"

_sensors, 2023, doi:10.3390/s23073658_

Round 1
Reviewer 1 Report
The research topic is interesting. Robust tracking of GNSS signals in challenging environments are the main concerned problem for these years. In this paper, the author proposed a complete single-frequency adaptive scalar tracking architecture named LBCA-based full LUT-DSKF. Equations are derived, and simulated experiments are performed. However, there are still some problems and revised suggestions for publication.
1.The main improvement of the proposed Full DSKF method compared to the method in reference [17] is the inclusion of the LBCA+DLL part. However, experimental results have shown that the LBCA FAP+DLL loop structure is the most optimal in terms of performance. The significance of this improvement is not clearly explained.
2. The effect of the LBCA algorithm on response time regulation was not demonstrated in the experiments. It is recommended to provide additional explanations and comparisons.
3.The keyword '(scalar tracking loop) SLT' does not appear in the abstract. It is suggested to remove it.
4. The A matrix in Equation 1 does not correspond to the text description 'The parameter υ determines the aiding of the carrier phase into the code phase'.
5. The role of the KF filter is not reflected in Figure 2. It is suggested to clarify the position of the KF filter in the structure, as well as its inputs and outputs.
6.On page 8, The statement 'the carrier phase and carrier Doppler difference, smoothed by the PLL and FLL loop filter, aids information to the code phase estimation' is inconsistent with elements A1,2 being zero in Equation 1 and Equation 14.
7.Figure 4 illustrates the relationship between loop bandwidth and weighting function, but the paper lacks a detailed derivation of the formulas.
8. The experimental setup lacks a comparison between the 'FAP+PAD' and 'FAP+DLL' loop structures and other loop structures.
9.Figure 12 illustrates the comparison between reference trajectory and estimated trajectory. However, detailed error analysis data is not described in the article.
Author Response
Please see the attachment "response_to_reviewer_1.pdf".

Reviewer 2 Report
The paper is practical, with a good theoretical and mathematical background on the discussed subject. Authors present and evaluate their own GNSS-purpose solution. However, in order to maintain high quality, Authors are advised to acquaint with the list of provided suggestions and comments, and make adequate modifications.
1) Consider discussing and presenting more content on positioning and navigation, with and without the aid of different algorithms, when it comes to mobile devices, i.e., smartphones, tablets, etc.
2) The matter of dual-band (dual-frequency) consumer electronics would be also interesting.
3) At the end of Chapter 1, close to the novelty highlights of this paper, Authors should clearly indicate which parts come from their previous Conference paper, which have been extended, and which are completely new added material.
4) Consider extending the number of cited references, the scope might include the abovementioned suggestions.
5) Will the GOOSE SBC cooperate with any kind of PC hardware and/or software, or is it only compatible with Windows (which version?) or Linux (which version?) distributions? What about required CPU power, RAM resources, etc.? At least basic comments seem necessary, referring to the tested laboratory stand.
6) What was the time duration of your studies? Did you monitor satellites during a single day, just for an hour? Or maybe the whole week? Additional comments seem necessary.
7) Where was the antenna mounted (indoor or outdoor environment)? What about its parameters and characteristics? What was the number of observed vs monitored satellites? Did it correlate with the theoretical trajectory of passing satellites (e.g., 90% accuracy, etc.)?
8) What was the surrounding environment, in which it was possible to maintain LOS conditions? Were there any time intervals with NLOS conditions?
9) Figures 10, 11 have a lot of details in it, therefore it would be better to present them in a column, rather than a row.
10) Authors could write more about open issues and future study directions.
Overall, it is a good or even a very good paper, it only requires some adjustments before it can be accepted and published in the Journal.
Author Response
Please see the attachment "response_to_reviewer_2.pdf".

Round 2
Reviewer 2 Report
The revised version of the manuscript has further improved its overall quality. Authors have adequately responded to my suggestions and comments. In my opinion, this paper is ready to be accepted and published.
This platform is very interesting, it seems to have a wide potential. Do continue your studies, including both simulation and field test measurements.